# NIH/3T3 Fibroblasts Selectively Activate T Cells Specific for Posttranslationally Modified Collagen Type II

**DOI:** 10.3390/ijms241310811

**Published:** 2023-06-28

**Authors:** Balik Dzhambazov, Tsvetelina Batsalova, Patrick Merky, Franziska Lange, Rikard Holmdahl

**Affiliations:** 1Faculty of Biology, Paisii Hilendarski University of Plovdiv, 4000 Plovdiv, Bulgaria; tsvetelina@uni-plovdiv.bg; 2Nykode Therapeutics ASA, 0349 Oslo, Norway; pmerky@gmail.com; 3Fraunhofer Institute for Cell Therapy and Immunology (IZI), 04103 Leipzig, Germany; franziska.lange@izi.fraunhofer.de; 4Section of Medical Inflammation Research, Department of Medical Biochemistry and Biophysics, Karolinska Institute, 17177 Stockholm, Sweden; rikard.holmdahl@ki.se

**Keywords:** fibroblasts, type II collagen (COL2), major histocompatibility complex class II (MHCII), T-cell hybridoma, rheumatoid arthritis

## Abstract

It has been shown that synovial fibroblasts (SF) play a key role in the initiation of inflammation and joint destruction, leading to arthritis progression. Fibroblasts may express major histocompatibility complex class II region (MHCII) molecules, and thus, they could be able to process and present antigens to immunocompetent cells. Here we examine whether different types of fibroblasts (synovial, dermal, and thymic murine fibroblasts, destructive LS48 fibroblasts, and noninvasive NIH/3T3 fibroblasts) may be involved in the initiation of rheumatoid arthritis (RA) pathogenesis and can process and present type II collagen (COL2)—an autoantigen associated with RA. Using a panel of MHCII/Aq-restricted T-cell hybridoma lines that specifically recognize an immunodominant COL2 epitope (COL2_259–273_), we found that NIH/3T3 fibroblasts activate several T-cell clones that recognize the posttranslationally glycosylated or hydroxylated COL2_259–273_ epitope. The HCQ.3 hybridoma, which is specific for the glycosylated immunodominant COL2 epitope 259–273 (Gal264), showed the strongest response. Interestingly, NIH/3T3 cells, but not destructive LS48 fibroblasts, synovial, dermal, or thymic fibroblasts, were able to stimulate the HCQ.3 hybridoma and other COL2-specific T-cell hybridomas. Our experiments revealed that NIH/3T3 fibroblasts are able to activate COL2-specific T-cell hybridomas even in the absence of COL2 or a posttranslationally modified COL2 peptide. The mechanism of this unusual activation is contact-dependent and involves the T-cell receptor (TCR) complex.

## 1. Introduction

Rheumatoid arthritis (RA) is a complex autoimmune disorder that generally affects peripheral joints and leads to significant physical disability, enhanced morbidity, and mortality [1,2]. During the disease course, the synovial tissue becomes infiltrated with immunocompetent cells and transforms into an invasive hypercellular structure called “pannus” that destroys the cartilage and erodes the bone. The main cell types in the pannus are macrophages, invasive synovial fibroblasts, and lymphocytes [3,4,5,6].

In RA pathogenesis, the intimal lining of the synovium changes dramatically from 1–2 cells deep to up to 10–20 cells. This hypercellularity results from an increase in the numbers of resident cell types in the structure—type A macrophage-like synoviocytes and type B fibroblast-like synoviocytes (also called synovial fibroblasts) [7]. Both cell types display an activated phenotype and produce proinflammatory cytokines, chemokines, growth factors, and destructive enzymes, which sustain inflammation, attract more cells to the joint, and contribute to extracellular matrix destruction.

Fibroblasts from different anatomic regions represent a diverse population of cells regarding their metabolic activities, proliferation, and expression of molecular markers and signaling molecules [8,9]. Activated fibroblasts are commonly involved in dynamic interactions with other cell types [4,10,11,12,13]. They express cytokines, specific enzymes, growth factors, lipid mediators, and numerous receptors, allowing them to respond to various signals. A role for fibroblasts in the development of periodontal diseases [14,15] or certain autoimmune diseases such as RA [16,17,18,19] and Graves’ disease [20,21] has been demonstrated. In fact, fibroblasts have been shown to react to danger signals and recruit immunocompetent cells, thus initiating inflammation and autoimmune reactivity in different tissues [22,23]. All these findings support the concept that they are active participants in immune reactions and contribute to the functioning of the professional immune system in health and disease.

Synovial fibroblasts (SF) play a key role in the pathogenesis of RA as the main contributors to the inflammatory process and joint destruction [3,10,18,24,25,26]. The autonomous pathogenic features of rheumatoid arthritis SF (RA SF) have been confirmed by in vivo experiments using mice with SCID (severe combined immunodeficiency) [27].

So far, several fibroblast subsets with distinct surface phenotypes have been identified [26]. The THY1 membrane glycoprotein (CD90) was used as a general marker for distinguishing two pathogenic fibroblast subsets related to the RA: THY1^+^ fibroblasts (located in the synovial sub-lining) that increase the severity and persistence of inflammation, and THY1^−^ fibroblasts (located in the synovial lining) that are responsible for cartilage and bone destruction [6,25,26,28,29].

Collagen type II (COL2) is the major molecule (90%) of the hyaline articular cartilage extracellular matrix [30,31,32,33,34]. It is a homotrimer of three α1(II) chains that contain high numbers of the repeating sequence Gly-Pro-X and is exclusively synthesized by chondrocytes [32,35]. Together with collagen types IX (COL9) and XI (COL11), COL2 forms a heteropolymeric fibrillar network, providing the main mechanical properties of the cartilage (strength and stiffness) and smooth surfaces of certain joints [32,33,36].

It has been shown that COL2 plays an important role in the pathogenesis of RA [37,38]. Immunization of experimental animals with COL2 leads to the development of collagen-induced arthritis (CIA), the most widely used model to study RA [39,40,41]. In both CIA and RA, the autoreactive T cells recognize the same COL2 binding motif (COL2_259–273_) [42,43,44], and many RA patients have anti-collagen antibodies [45,46,47,48].

Susceptibility to CIA in mice is associated with specific major histocompatibility complex class II (MHCII) haplotypes (I-A^q^ and I-A^r^) [49,50]. Similarly, RA susceptibility in patients is associated with human leukocyte antigens (HLA-DRB1) molecules (shared epitope hypothesis) [51,52], which present self-antigens to autoreactive T cells. Both mouse and human MHCII molecules bind and present the immunodominant epitope COL2_259–273_ to the T cells [53,54,55]. The extracellular matrix of the joints is a rich source of autoantigens, including COL2. Thus, the T-cell response to COL2 is restricted by MHCII, suggesting a key role for antigen-presenting cells (APCs) in the pathogenesis of RA.

In addition to their ability to destroy joint tissue directly, synovial fibroblasts can also act as APCs and eventually stimulate pathogenic T lymphocytes [56]. It has been shown that SF can activate MHCII-restricted T cell hybridomas specific for arthritogenic peptides [57]. In fact, the APC properties of interferon gamma (IFN-γ)-pulsed SF have been confirmed by several research groups, while other types of fibroblasts, i.e., dermal fibroblasts, show defects in their APC capacity [57,58,59,60]. Based on these findings, our research aimed to further clarify the role of fibroblasts in the initiation of the RA pathogenesis, similarly to the professional APCs, and examine whether different types of fibroblasts could process and present the main cartilage protein involved in the pathogenesis of RA—COL2 [61]—to MHCII (A^q^)-restricted COL2-specific T-cell hybridomas. For these experiments, we used normal murine synovial fibroblasts, unaffected by an inflammatory milieu, dermal and thymic mouse fibroblasts, a murine fibroblast cell line LS48 with a well-defined ability to invade and destruct cartilage both in vitro and in vivo [62,63], and a non-invasive murine fibroblast line NIH/3T3 [63,64]. Interestingly, only NIH/3T3 fibroblasts were able to stimulate T-cell hybridoma clones specific for the glycosylated or hydroxylated immunodominant COL2 peptide. The mechanism of this unconventional activation, which occurs even in the absence of COL2 peptides, was investigated.

## 2. Results

### 2.1. Destructive LS48 Fibroblasts, Synovial/Dermal and Thymic Fibroblasts Cannot Process and Present COL2 to T-Cells

To determine the antigen-presenting capacity of different types of fibroblasts to MHCII (A^q^)-restricted COL2-specific T-cell hybridomas, we used isolated murine thymic fibroblasts (ThyF), synovial fibroblasts (SF), and dermal fibroblasts (DF), as well as the cell line NIH/3T3. All of these are derived from mouse strains with the MHCII H2^q^ haplotype, thus expressing A^q^. In addition, we used destructive LS48 fibroblasts. To determine whether the fibroblast cell lines (NIH/3T3 and LS48) and to confirm that isolated thymic fibroblasts from B10.Q mice express particularly A^q^ molecules, we have performed fluorescent staining by using the antibody KH116, which is specific for this molecule (Figure 1). The microscopic analysis confirmed that NIH/3T3 cells (Figure 1A) and thymic fibroblasts (Figure 1B) express A^q^, while the LS48 cells showed negative results (Figure 1C). Although dermal and synovial fibroblasts were isolated from QB mice, which have I-A^q^ haplotypes of MHCII, we have checked again the expression of I-A^q^ molecules. The results are presented in the Appendix A.

Each fibroblastic type was co-cultured with the T-cell hybridoma HCQ.3, which specifically recognizes COL2 and the glycosylated COL2 immunodominant epitope 259–273 (Gal259–273) bound to the A^q^ molecule. The cells were incubated in the presence of antigen (50 µg/mL COL2) for 24 h and then the IL-2 level in the culture medium was measured in order to assess activation of the T-cell hybridoma [55,65]. Interestingly, only when co-cultured with NIH/3T3 murine fibroblasts, the HCQ.3 T cells produced a significant amount of IL-2 (Figure 2). Neither the isolated thymic, synovial, or dermal mouse fibroblasts nor the destructive fibroblastic cell line LS48 activated the HCQ.3 T cell hybridoma in the presence of COL2 protein. Results from the performed tests related to the antigen-presenting capacity of the fibroblasts to other COL2-specific T cell hybridomas are presented in the Appendix A.

### 2.2. NIH/3T3 Cells Are Able to Stimulate the COL2-Specific HCQ.3 Hybridoma Even in the Absence of Antigen

To confirm and further characterize the antigen-presenting capacity of NIH/3T3 fibroblasts, we performed co-culture experiments with additional controls, and the cells were incubated in the presence of either the COL2 protein or the Gal259–273 peptide (Figure 3). Control samples contained only fibroblasts or hybridomas, fibroblasts and hybridomas without antigen, as well as fibroblasts, hybridomas, antigen (COL2 or Gal259–273), and APCs (1 × 10^6^ murine splenocytes/well). As shown in Figure 3, NIH/3T3 cells elicited strong responses from the HCQ.3 hybridoma independent of the presence of APCs (Figure 3A). ThyF was co-cultured with the Gal259–273 peptide and HCQ.3 cells generated a very weak hybridoma response (Figure 3B), which suggests that ThyF are not able to process COL2 protein (Figure 2) but can present a short COL2 peptide to the T-cell hybridoma HCQ.3. LS48 and QB DF did not induce T-cell hybridoma activation—IL-2 production was detected only in the control samples containing fibroblasts, T-cell hybridoma, antigen, and APCs (Figure 3C,D). However, the most intriguing result from this experiment came from one of the control samples. Significant IL-2 production was detected in the culture wells containing only NIH/3T3 fibroblasts and HCQ.3 cells without COL2 or Gal259–273. This activation in the absence of antigen suggests that NIH/3T3 cells do not act as APCs but rather stimulate the T-cell hybridoma by a novel, unconventional mechanism. The results gained from this experiment showed that non-activated normal fibroblasts are not able to process and present COL2 to T-cells, suggesting that the fibroblasts are not participating as APCs in the pathogenesis of RA. On the other hand, the activation of COL2 specific T-cells by NIH/3T3 fibroblasts without antigen directed our study to the unusual antigen-independent mechanism of T-cell hybridoma activation.

### 2.3. NIH/3T3 Fibroblasts Selectively Activate the COL2-Specific T-Cell Hybridoma HCQ.3

To examine whether NIH/3T3 fibroblasts can be stimulated without COL2 antigens, different T-cell hybridoma clones, specific for COL2, non-modified, or posttranslationally modified immunodominant COL2_259–273_ epitopes, were used for co-culture experiments. NIH/3T3 cells were co-cultured with HCQ.2, HCQ.3, HCQ.4, HCQ.6, HCQ.10, HCQ.11, 22a1.7E, HRC.2, HDQ.9, or HDBR.1 clones for 24 h and subsequently, IL-2 content was measured in culture medium samples to evaluate eventual hybridoma activation. The results from this experiment are presented in Figure 4A.

Interestingly, only hybridomas specific for posttranslational modifications (HCQ3, 22a1.7E, HCQ.10, HCQ.11, and HDBR.1) showed increased IL-2 levels. The non-responding clones were specific for a non-modified peptide (HCQ.4, HRC.2, HDQ.9), whereas two of them (HCQ.2 and HCQ.6) are specific for a posttranslationally modified peptide (Gal264). Moreover, other fibroblastic types (Figure 4B) could not induce an IL-2 response in HCQ.3 cells following 24 h co-culture. One could speculate that this unusual stimulation is an artifact caused by the fusion partner used to generate the hybridoma. BW5147-specific molecules expressed on the surface of HCQ.3 could interact with NIH/3T3 and hence induce hybridoma activation. To investigate this assumption, NIH/3T3 fibroblasts, as well as other types of fibroblasts, were co-cultured for 24 h with the TCR-negative BW5147 fusion partner cells. However, a significant IL-2 level was not detected in the co-culture supernatant after the incubation period (Figure 4C). This result excludes the possible role of fusion partner molecules in the selective activation of HCQ.3 cells by NIH/3T3 fibroblasts.

### 2.4. The Mechanism of T-Cell Hybridoma Activation by NIH/3T3 Fibroblasts Is Contact-Dependent

The next step in our investigation was to determine the basic mechanism of T-cell hybridoma activation by NIH/3T3—whether it depends on direct cell-cell contact or on soluble mediators secreted by the cells. For this purpose, NIH/3T3 and HCQ.3 cells were co-cultured in microtiter plates with transwell membranes that prevent direct physical contact between the hybridomas and the fibroblasts. HCQ.11, one of the clones that showed increased IL-2 production after incubation with NIH/3T3 fibroblasts (Figure 4A), was also included in the experiment. As an additional control, HRC.2 cells, which did not show activation in co-culture with NIH/3T3 (Figure 4A), were used.

The results we gained undoubtedly confirmed that the mechanism of interaction between the COL2-specific T-cell hybridomas and NIH/3T3 cells is cell-cell contact-dependent. HCQ.3, as well as HCQ.11 cells, did not produce a significant amount of IL-2 when co-cultured separated by a transwell membrane with NIH/3T3 fibroblasts (Figure 5).

Both T-cell hybridomas, HCQ.3 and HCQ.11, were activated only when co-cultured with NIH/3T3 without a transwell membrane. In addition, the levels of TNF-α, IL-6, IL-10, and IFN-β were measured in all test samples. Significant production of these cytokines was not detected, proving further the contact-dependent pattern of interaction between the cells.

### 2.5. IFN-Gamma Down-Regulates the Expression of MHCII and CD154 on NIH/3T3 Fibroblasts, but Up-Regulates the Expression of These Markers on LS48 Destructive Fibroblasts and QB Synovial Fibroblasts

It has been shown that IFN-γ up-regulates the expression of MHCII molecules on fibroblasts, and IFN-γ-pulsed RA SF effectively presents arthritogenic peptides to T-cell hybridomas [57,60,66]. Thus, to assess the effect of such stimulation on NIH/3T3 fibroblasts, QB synovial fibroblasts and destructive LS48 fibroblasts were treated for 72 h with 100 U/mL IFN-γ and then the expression of MHCII A^q^ molecules was examined by flow cytometry. An analysis of CD154 expression was also performed. CD154 (also known as CD40L) is a surface molecule that can be expressed by fibroblasts and is considered to play a role in fibroblast activation [8,67]. Interestingly, IFN-γ treatment had a reversed effect on NIH/3T3 cells compared to other types of fibroblasts: IFN-γ-pulsed NIH/3T3 cells showed reduced expression of MHCII molecules and CD154 (Figure 6A,B). On the other hand, QB synovial fibroblasts and LS48 cells up-regulated their expression of MHCII and CD154 following IFN-γ treatment.

It has been shown that NIH/3T3 as well as other types of fibroblasts express both CD40L and CD40 molecules. Ligation of CD40 induces SF proliferation and up-regulation of adhesion molecules such as ICAM-1 and VCAM-1 [68,69]. To evaluate the role of MHCII, CD40, and CD40L on NIH/3T3 cells for T-cell hybridoma activation, the molecules were blocked with monoclonal antibodies. After that, antibody-treated NIH/3T3 fibroblasts were co-cultured with HCQ.3 cells, and the level of hybridoma activation was analyzed. Figure 6C represents the results of this experiment. Blocking of MHCII, CD40, or CD40L did not affect the interaction between NIH/3T3 and T-cell hybridoma cells. Moreover, it showed a stimulatory effect—fibroblasts with blocked MHCII, CD40, or CD40L surface molecules elicited a higher level of IL-2 production compared to untreated NIH/3T3 cells.

### 2.6. The Interaction between NIH/3T3 Fibroblasts and HCQ.3 Cells Involves the TCR Complex

After determining the contact-dependent nature of COL2-specific T-cell hybridoma activation by NIH/3T3 cells, our studies aimed to determine which cell surface molecules are major players in this interaction. Using specific monoclonal antibodies, different surface molecules (CD3, CD4, TCR, CD34, CD44, CD55, CD90.2, CD106, and CD138), associated with fibroblast phenotype, activity, and adhesion, were blocked. As shown in Figure 7A, blocking adhesion molecules CD34 and CD106 had a stimulatory effect, while blocking CD44, CD55, CD90.2, and CD138 did not influence the interaction between NIH/3T3 and T-cell hybridoma cells (Figure 7B). Most importantly, blocking TCR, CD3, and CD4 co-stimulatory molecules on HCQ.3 cells inhibited their activation following co-culture with NIH/3T3 cells (Figure 7A). These results demonstrate that the mechanism of COL2-specific hybridoma activation by NIH/3T3 fibroblasts involves the TCR complex.

## 3. Discussion

Fibroblasts comprise a heterogeneous population of cells with diverse structure, metabolic, and functional activities [8,25,70,71,72,73]. They are now considered a part of the professional immune system because of their diverse expression of immunomodulatory factors such as cytokines, growth factors, and specific surface and intracellular receptors that enable them to react to danger signals, recruit immunocompetent cells, and contribute to tissue remodeling and wound healing.

It is well documented that fibroblasts express MHCII molecules, specific adhesion molecules, and cytokines, and therefore it has been hypothesized that fibroblasts could act as antigen-presenting cells. Indeed, several reports have shown the antigen-presenting ability of fibroblasts [57,58,60]. It has been reported that synovial fibroblasts are able to present arthritogenic peptides. However, to date, there is no clear evidence for the ability of fibroblasts to take up and process antigens with the same efficiency as professional APCs. Therefore, in relation to RA, our investigation aimed to clarify if fibroblasts can process and present COL2.

Previous research on the antigen-presenting capacity of fibroblasts has shown various results. Kundig et al. demonstrated that fibroblasts in lymphoid organs are able to present antigens in vivo [74]. Dermal fibroblasts were shown to effectively take up and process antigens, but when treated with IFN-γ they failed to present antigens [59]. Gingival fibroblasts also failed to present antigens derived from periodontopathic bacteria [75]. Periodontal ligament fibroblasts were shown to express MHCII molecules, but instead of presenting antigens, they rather acted as receptor molecules that transmit signals into fibroblasts and induce cytokine production [60]. This is of extreme importance in the selection of therapeutic options for immunoregulation and control of the inflammatory process, especially when complex systemic therapies are applied [76].

It has been demonstrated that synovial fibroblasts can process and present exogenous bacterial antigens to T-cell clones and peripheral blood lymphocytes via MHC-restricted mechanisms [58,77]. Recent research showed that SF can take up and present arthritogenic HCgp39 and COL2 peptides, but no evidence for their ability to process autoantigens was found [57]. Here we show for the first time an investigation on the capacity of different types of fibroblasts to process COL2 and present COL2 antigens to murine T-cell clones. Five types of murine fibroblasts were used in the present study: embryonic NIH/3T3 fibroblasts, destructive LS48 fibroblasts, primary QB synovial fibroblasts, primary QB dermal fibroblasts, and primary thymic fibroblasts. Among them, only NIH/3T3 cells were able to induce responses in COL2-reactive T-cell hybridomas. In particular, activation was detected for the hybridomas that specifically recognize the posttranslationally glycosylated or hydroxylated COL2_259–273_ immunodominant epitope. Strikingly, further experiments revealed that NIH/3T3 cells can activate T-cell hybridomas even without COL2 antigens. The clone HCQ.3 that showed the highest response in co-culture with NIH/3T3 cells is specific for the glycosylated COL2_259–273_ epitope. This unusual finding confuted our initial hypothesis for effective APC function in NIH/3T3 fibroblasts and directed the aim of our research toward an investigation of the mechanism of antigen-independent T-cell activation by NIH/3T3. Thereafter, we determined that the mechanism of T-cell hybridoma activation is cell-cell contact-dependent because co-culture of NIH/3T3 fibroblasts and HCQ.3 cells in transwell membrane plates that prevent direct physical contact between the two cell types did not elicit an IL-2 response by the T-cell hybridomas. In addition, no significant cytokine production, i.e., IL-6, IL-10, TNF-α, or IFN-β production, was detected. Similar results were reported by Mori et al., showing that synovial fibroblast-mediated activation of CD4+ T-cells is cell-cell contact-dependent but not dependent on soluble molecules [78]. Gjertsson et al. Showed that A_a_^p^/A_b_^q+^ NIH/3T3 cells transduced with LNT-Ii-COL2 presented both non-modified and glycosylated forms of the COL2 peptide to the T-cell hybridomas, which recognize different glycosylation patterns of the COL2 peptide (including HCQ.3), although the LNT-Ii-COL2 transgene is only for the non-modified form of the COL2 peptide [79]. Most probably, the response of the T-cell hybridomas specific for glycosylated forms of COL2 is due more to the contact-dependent activation described here than to the LNT-Ii-COL2 transgene followed by its glycosylation.

Our next studies aim to identify which cell surface molecules are key players in the NIH/3T3-HCQ.3 hybridoma interaction. Among the cell-cell contact-dependent interactions leading to T-cell activation, a role for CD154 (CD40L)-CD40 engagement has been highlighted. It has been shown that CD40 is expressed on synovial fibroblasts and destructive murine fibroblasts [80,81], and ligation of CD40 induces SF proliferation and up-regulation of adhesion molecules [68,69]. However, CD154 expression on T-cells is transient and is rapidly down-regulated after binding to CD40 [82]. Prominent T-cell activation, as seen in the experiments with HCQ.3 hybridoma, probably could not be induced by this short-term interaction. On the other hand, it was shown that fibroblasts express CD154, and dysregulated C154 expression could play a role in chronic cell activation leading to disease [67]. Based on these data, we decided to test the importance of both CD40 and CD154 for the COL2-independent activation of COL2-specific T-cell hybridomas by NIH/3T3 cells. However, blocking CD40 or CD154 did not affect the interaction between HCQ.3 and NIH/3T3, suggesting that other surface molecules mediate this unusual T-cell stimulation.

Other classes of cell surface molecules on NIH/3T3 cells that were targeted and blocked with monoclonal antibodies include the adhesion molecules CD44 and CD106. CD44 has been associated with RA pathogenesis [80]. However, no effect was evident following the blockage of this marker. Another important adhesion molecule chosen for blocking experiments is CD106 because it is the predominant adhesion molecule on SF in the inflamed synovium [83] and also because it is involved in T-cell recruitment and co-stimulation [80]. The expression of CD106 is significantly higher in NIH/3T3 fibroblasts compared to destructive LS48 [80], which is another important reason to investigate the role of this molecule for selective T-cell activation by NIH/3T3. Nonetheless, blocking of CD106, similar to blocking of CD44, did not yield a negative effect on HCQ.3 T-cell hybridoma activation upon co-culture with NIH/3T3. On the contrary, CD44 and CD106 had even more stimulatory effects on the HCQ.3-NIH/3T3 interaction.

The CD90 molecule and its isoform CD90.2 are fibroblast-specific cell surface molecules [84], as is syndecan-1 (CD138), a proteoglycan involved in numerous cell functions. Another line of investigation on the mechanism of selective T-cell activation by NIH/3T3 cells concentrated on fibroblast-specific markers. Both CD90.2 and CD138 are heavily glycosylated molecules, which could also contribute to T-cell activation by mimicking the COL2-glycosylated epitope that they recognize. Again, however, blocking either CD90.2 or CD138 did not affect the investigated interaction.

In line with testing specific cell surface molecules and immunological markers, the role of two classic molecules involved in T-cell activation was investigated: TCR and MHCII. Interestingly, treatment of HCQ.3 T-cell hybridoma with anti-TCR antibodies resulted in a tremendous reduction of IL-2 production, demonstrating that the TCR is the major cell surface molecule complex involved in HCQ.3 activation even in the absence of antigen. Blocking of MHCII molecules, on the other hand, resulted in increased IL-2 production by COL2-specific T-cell hybridomas, suggesting that the TCR recognizes a non-COL2 glycosylated or hydroxylated epitope on NIH/3T3 cells (aside from the MHC class II complex). Probably a glycosylated or hydroxylated molecule on NIH/3T3 fibroblasts mimics the COL2 epitope and stimulates T-cell hybridomas specific for these posttranslationally modified epitopes. Our findings suggest the existence of another mechanism by which bystander cells may be involved in the pathogenesis of RA, namely through direct contact with arthritogenic T cells, activating them by using molecules that mimic the relevant antigens. Strikingly, this activation depends on CD4 and CD3 co-receptor molecules without specific presentation on MHCII in NIH/3T3 cells. It is very likely that this interaction between NIH/3T3 cells and T cells and their subsequent activation is mediated by galectins, one of the most interesting and studied classes of lectins. The detailed mechanism of this unusual interaction remains to be revealed.

Our study has some limitations. The present study included fibroblasts that were not affected by inflammatory processes, as our idea was to test whether non-activated fibroblasts could have the potential to initiate the development of RA by processing and presenting the major structural element of articular cartilage and the candidate autoantigen COL2, similarly to professional antigen-presenting cells. We found that different types of fibroblasts could not process and present COL2 to T-cell hybridomas under optimal conditions for hybridoma assays. It is possible that under conditions of joint inflammation, secreted IFN-γ activates synovial fibroblasts, increasing MHCII expression, and, in the presence of COL2 fragments and infiltrated COL2-specific T cells, activates fibroblasts and functions as APCs. Although this is another situation that is related not to the initiation but to the development of RA, it is still possible that synovial fibroblasts fulfill the role of APCs, and this depends on both the concentration of the released COL2 fragments and the duration of interaction with the infiltrating arthritogenic T-cells. Furthermore, our finding that NIH/3T3 fibroblasts can activate specific T-cell clones that recognize posttranslationally modified COL2 epitopes cannot be extrapolated to all fibroblasts because the studied primary cultures of fibroblasts isolated from different tissues and expressing correct MHCII molecules (A^q^) showed no such activity. On the other side, the fact that till now only NIH/3T3 fibroblasts can activate COL2-specific T-cells and that this activation is MHCII-restricted and contact-dependent suggests that the initiation of RA pathogenesis might be possible with the participation of mimicking antigens and bystander cells. Further studies are needed to verify whether, under conditions of inflammation, synovial fibroblasts can perform the function of antigen-presenting cells and whether, under such conditions, they express glycoproteins that mimic COL2 epitopes and can activate COL2-specific T-cells.

## 4. Materials and Methods

### 4.1. Cell Lines

#### 4.1.1. T-Cell Hybridomas and Fine Specificities

Ten COL2-specific T-cell hybridoma lines were used in the present study: HCQ.2, HCQ.3, HCQ.4, HCQ.6, HCQ.10, HCQ.11, HDBR.1, 22a1.7E, HRC.2, HDQ.9. They were generated as previously described by Corthay et al. [65,85]. All HCQ hybridomas were derived from COL2-immunized C3H.Q mice, and one was from COL2-immunized DBA/1 mice. The TCR-negative BW5147 cell line [86] was used as a fusion partner for all COL2-stimulated T-cells, isolated from popliteal and inguinal lymph nodes of COL2-immunized C3H and DBA/1 mice.

The fine specificity of the used hybridomas is presented in Table 1. HRC.2, HDQ.9, and HCQ.4 hybridoma clones specifically recognize COL2 and the non-modified COL2_259–273_ epitope (naked peptide). The HDBR.1 clone is specific for COL2 and the posttranslationally modified COL2_259–273_ epitope containing hydroxylysine at position 264 (Hyl264 peptide). The 22a1.7E, HCQ.3, and HCQ.6 hybridomas are specific for COL2 and the Gal264 peptide (posttranslationally modified COL2_259–273_ epitope with a glycosylated lysine at position 264). The HCQ.2 hybridoma recognizes the COL2 molecule and both the Gal264 peptide and the Gal270 peptide (the COL2_259–273_ epitope, which contains posttranslationally glycosylated lysine residues at positions 264 and 270). The HCQ.10 hybridoma recognizes the COL2 protein, the Gal264 peptide, and the GalHnv264/270 peptide (a peptide that corresponds to COL2_259–273_ but is modified at positions 264 and 270 with glycosylated 5-hydroxy-L-norvaline—an artificial molecule that lacks the aminomethylene group of the natural lysine residue). The HCQ.11 T-cell hybridoma clone recognizes the COL2 protein and the posttranslationally modified COL2_259–273_ peptide with both Gal and Glu on position 264 (the Glc-gal peptide) [65].

#### 4.1.2. LS48 Cell Line

The cell line LS48 was a generous gift by Prof. Ulrich Sack, Leipzig University, Leipzig, Germany. This continuous murine cell line [German collection of micro-organisms and cell cultures Braunschweig (DSMZ)—DSM ACC 2455; Biotectid, Leipzig, Germany] has been established from a co-culture of mouse peritoneal fibroblasts (from SCID mice) and human rheumatoid synovial fibroblasts [63]. The cells were co-cultured with repeated passage for a period of 12 weeks until the human cells were no longer detectable.

#### 4.1.3. NIH/3T3 Cell Line

NIH/3T3 (American Type Culture Collection, Rockville, MD, USA, ATCC^®^ CRL-1658™) is a continuous fibroblast cell line derived from NIH/Swiss mouse embryo cultures by the same method as the original random-bred 3T3 (ATCC^®^ CCL-92™) and the inbred BALB/c 3T3 (ATCC^®^ CCL-163™) [64]. The NIH/3T3 cells are highly sensitive to sarcoma virus focus formation and leukemia virus propagation. In confluent cultures, NIH/3T3 proliferation is contact-inhibited.

### 4.2. Fluorescence Microscopy

The expression of I-A^q^ on the NIH/3T3, thymic fibroblasts, and LS48 cells was analyzed by fluorescence microscopy using a monoclonal biotin mouse anti-mouse I-A(q) antibody (clone KH116, BD Biosciences, San Jose, CA, USA) and FITC Streptavidin (BD Biosciences, San Jose, CA, USA). Cells were seeded on coverslips in 12-well plates (1 × 10^5^ cells/well) and cultured in a humidified incubator at 37 °C with 5% CO_2_ in Dulbecco’s Modified Eagle Medium, DMEM (1 mL/well) supplemented with 10% fetal calf serum (FCS) and antibiotics (100 IU/mL penicillin and 100 μg/mL streptomycin). After 24 h, the coverslips with the adhered cells were washed with phosphate buffered saline (PBS) and stained with KH116/FITC Streptavidin for 15 min at 37 °C in the dark. After washing three times with PBS, the cells were analyzed with a fluorescent microscope (Leica Microsystems GmbH, Wetzlar, Germany) equipped with a camera.

### 4.3. Primary Murine Fibroblasts

All primary mouse fibroblasts were previously isolated (at Medical Inflammation Research, Lund University, ethical permit number M70-04), and cells from the third passage were frozen and stored in liquid nitrogen until use. Synovial and thymic fibroblasts were isolated from adult F1(B10.QxBalb/c) (QB) or B10.Q mice. Dermal fibroblasts were isolated from the skin of newborn QB or B10.Q mice. Fibroblasts were isolated from these mouse strains to be sure that they expressed I-A^q^ molecules and could present antigens to A^q^ restricted T-cell hybridomas. In brief, the mice were sacrificed, and thymi, synovial tissue, or skin fragments were dissected and transferred to a tube with sterile D-PBS (Gibco^®^, Life Technologies™, Paisley, UK) supplemented with 100 IU/mL penicillin, 100 µg/mL streptomycin, and 2.5 µg/mL amphotericin B (SERVA, Heidelberg, Germany). All samples were minced and digested with 0.1% trypsin in PBS for 45 min. at 37 °C. Then, the samples were additionally digested by incubation in DMEM (Gibco^®^, Life Technologies™, Paisley, UK) containing 0.1% collagenase D (Roche Diagnostics GmbH, Mannheim, Germany) and 10% heat-inactivated fetal calf serum (FCS) (PAA Laboratories GmbH, Linz, Austria) for 90 min at 37 °C. After that, single-cell suspensions were derived by passing the samples through a 40 µm cell strainer (FALCON^®^, Becton Dickinson, Le Pont De Claix, France). The cells were washed twice in DMEM and cultured as described in the next section. Contamination with other cells was eliminated following three to four passages of the cultures, and the phenotype of the fibroblasts was confirmed by flow cytometry using CD106 (VCAM-1), CD68, and CD11b as markers to exclude contamination with other cell types.

### 4.4. Cell Culture Conditions

The continuous cell lines and primary murine fibroblasts were kept in a humidified incubator maintaining 37 °C and cultured in complete DMEM—Dulbecco’s Modified Eagle Medium (Gibco^®^, Life Technologies™, Paisley, UK) supplemented with 10% heat-inactivated FCS (PAA Laboratories GmbH, Linz, Austria) and antibiotics (100 IU/mL penicillin, 100 µg/mL streptomycin) (Merck KGaA, Darmstadt, Germany). The hybridoma cells were cultured in 25-cm^2^ flasks. The primary murine fibroblasts, LS48 and NIH/3T3 cells, were grown in 25- and 75-cm^2^ culture flasks (TPP, Trasadingen, Switzerland). When LS48, NIH/3T3, and primary fibroblast cultures reached confluence, the cells were detached by incubation with a 0.25% trypsin/0.53 mM ethylenediaminetetraacetic acid (EDTA) solution for 3–5 min, diluted 1:10 in complete DMEM, and placed in new 75-cm^2^ culture flasks. When hybridoma cultures reached stationary phase, the cells were thoroughly resuspended, diluted 1:10 in complete DMEM, and transferred to new 25-cm^2^ flasks.

### 4.5. Co-Culture Experiments

To determine their antigen-presenting capacity, NIH/3T3, LS48, dermal, thymic, or synovial murine fibroblasts were co-cultured with A^q^-restricted COL2-specific T-cell hybridomas in the presence of COL2 protein, COL2 peptides, or other cartilage-specific proteins. For this assay, we used a previously described technique [65]: T-cell hybridomas (5 × 10^4^ cells/well) were incubated with fibroblasts (5 × 10^5^ cells/well) and protein or peptide antigen in a total volume of 200 µL in 96-well microtiter flat-bottom plates (TPP, Trasadingen, Switzerland). After 24 h, 100 µL of the culture supernatant were removed, frozen, and used consequently for the determination of IL-2 content. Analogously, the co-culture assay was performed using Corning^®^ HTS Transwell^®^ 96 well plates with a 0.4 µm pore polycarbonate membrane (Merck KgaA, Darmstadt, Germany) to determine whether the interaction between fibroblasts and T-cell hybridomas is contact-dependent.

### 4.6. Enzyme-Linked Immunosorbent Assay (ELISA)

The IL-2 content in culture supernatants was measured by sandwich ELISA. Microtiter plates (Corning—Life Sciences, Kennebunk, ME, USA) were coated with 5 µg/mL anti-mouse IL-2 antibody (clone Jes6IA12) for 2 h at room temperature. After that, the plates were washed with ELISA buffer (PBS buffer + 0.05% Tween 20), and unspecific binding was blocked by a 30 min incubation with a 2% solution of non-fat milk. Subsequently, the plates were washed with an ELISA buffer, and 50 µL of culture supernatant was added to the plates and incubated for 2 h at room temperature. At the end of the incubation period, the plates were washed and incubated for 1 h with a 50 µL/well biotinylated anti-mouse IL-2 antibody solution (1 µg/mL concentration; clone 5H4). Specifically bound IL-2 was detected using Eu^3+^-labeled streptavidine and the DELFIA system (PerkinElmer, Waltham, MA, USA). The plates were analyzed on a Wallac multi-label reader (Wallac Finland Oy, Turku, Finland). Recombinant mouse IL-2 served as a positive control and standard. All samples were plated in triplicate.

### 4.7. Blocking of Cell Surface Molecules

NIH/3T3 fibroblasts were treated with different monoclonal antibodies in order to block specific cell surface molecules. The cells were trypsinized, and 1 × 10^6^ cells of the resulting suspension were incubated for 1 h at 37 °C with 3 µg/mL anti-mouse MHC class II, CD34, CD40, CD40L, CD44, CD55, CD90.2, CD138, or CD106 antibodies (BD Biosciences, San José, CA, USA; BioLegend, San Diego, CA, USA). Following treatment, NIH/3T3 cells were washed with PBS, resuspended in complete DMEM, and then plated in co-culture with T-cell hybridomas for 24 h. Thereafter, 100 µL co-culture supernatant was collected and assayed for IL-2 content by ELISA.

Cell surface molecules on HCQ.3 cells were blocked with monoclonal anti-mouse CD3, CD4, or TCR Vβ8 chains. The antibodies were added to the cell culture medium to a final concentration of 3 µg/mL and incubated with 1 × 10^6^ hybridomas for 1 h at 37 °C. Then the cells were washed with PBS, resuspended in complete DMEM, and seeded in co-culture with NIH/3T3 fibroblasts. After 24 h 100 µL of co-culture supernatant was collected and analyzed for IL-2 content by ELISA.

### 4.8. Flow Cytometry

IFN-γ-pulsed and untreated fibroblasts were analyzed by flow cytometry. Synovial QB fibroblasts, LS48, and NIH/3T3 cells were incubated for 72 h in complete DMEM containing 100 U/mL IFN-γ. The corresponding control cells were cultured in standard, complete DMEM. At the end of the incubation period, LS48, NIH/3T3, and QB fibroblasts were detached from the culture vessels using trypsin-EDTA solution and washed twice with D-PBS (Gibco^®^, Life Technologies™, Paisley, Scotland, UK) supplemented with 1% FCS (PAA Laboratories GmbH, Linz, Austria). Anti-FcRIIε antibody (clone 2.4.G2 from our antibody collection) was used prior to staining with the aim of blocking unspecific binding to Fc receptors. After manual counting, 1 × 10^5^ cells were resuspended in 100 µL D-PBS with 1% FCS and stained for MHC class II, CD40L, CD62L, CD68, CD106, and CD11b surface molecules using fluorochrome-labeled antibodies (BioLegend, San Diego, CA, USA; BD Biosciences, San José, CA, USA). The control and IFN-γ-treated cells were incubated with the corresponding antibodies for 15 min at room temperature in the dark. After that, the cells were washed twice with D-PBS + 1% FCS and analyzed with a flow cytometer (LSR II, BD Biosciences, San José, CA, USA). Debris and dead cells were excluded from the evaluation by common gating procedures. Flow cytometry data was analyzed using FlowJo software Version 7.8.2 (FlowJo LLC, Ashland, OR, USA).

### 4.9. Statistics

Data are presented as the mean ± standard error of the mean (SEM). A non-parametric Mann-Whitney U test was applied for statistical analysis using the StatView software Version 5.0 (SAS Institute Inc., Cary, NC, USA). The differences were compared with the corresponding controls. *p* values less than 0.05 were considered statistically significant.

## 5. Conclusions

Here, we show for the first time that in the absence of an inflammatory milieu, different types of murine fibroblasts are not able to process and present COL2 to arthritogenic T-cells and that NIH/3T3 mouse embryonic fibroblasts stimulate COL2-specific T-cell hybridomas in the absence of antigen using an unconventional TCR complex-dependent mechanism. Thus, a direct contact-dependent mechanism based on the interaction between autoantigen-mimicking molecules on the surface of non-professional APCs and T cells may be involved in the initiation of RA pathogenesis.

On the other hand, under normal physiological conditions and especially in the presence of an inflammatory process, the interaction between infiltrated T lymphocytes and synovial fibroblasts or other bystander cells is much longer, and when collagen fragments accumulate, their processing and presentation by synovial fibroblasts are not excluded. Further studies are needed to elucidate the role of the concentration of collagen fragments as well as the duration of interaction between synovial fibroblasts and COL2-specific T-cells in the initiation of an immune response and the development of RA.

The data from the present study also suggest the need for a precise selection of therapeutic methods to regulate the immune response, taking into account the possible causes of its invocation.

## Figures and Tables

**Figure 1 ijms-24-10811-f001:**
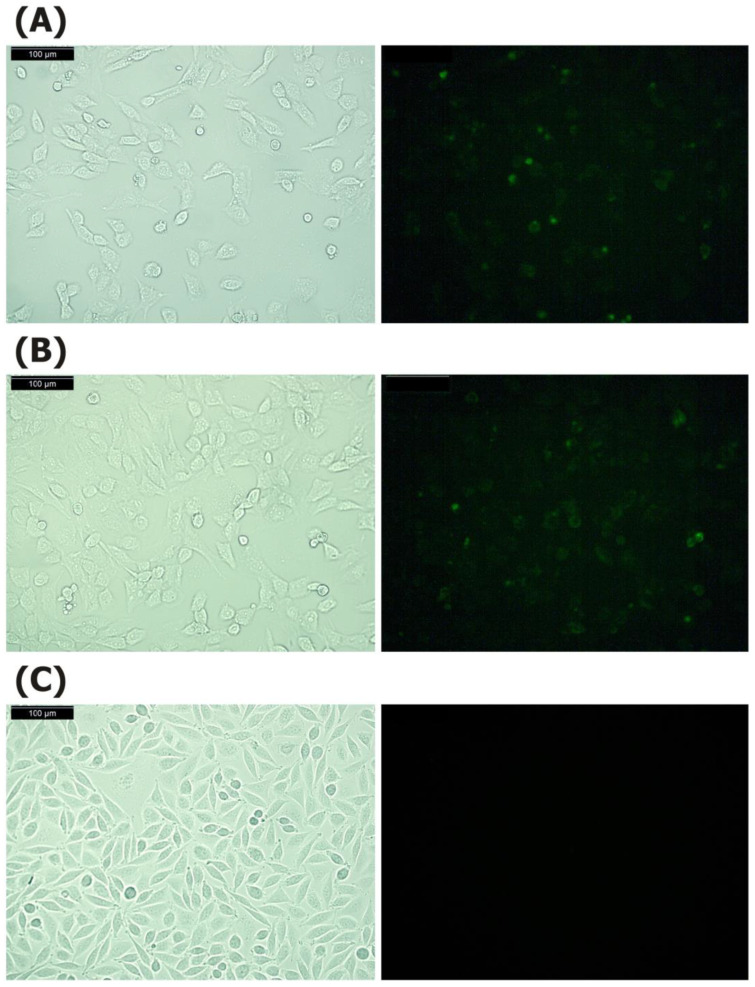
Expression of the A^q^ molecule on NIH/3T3 cells (**A**), thymic fibroblasts (**B**) and LS48 cells (**C**). After adhesion for 24 h, the cells were stained with biotin labeled anti-mouse A^q^ (clone KH116) and FITC Streptavidin for 15 min at 37 °C in dark. Photomicrographs show light/fluorescence from the respective area. Bar 100 µm.

**Figure 2 ijms-24-10811-f002:**
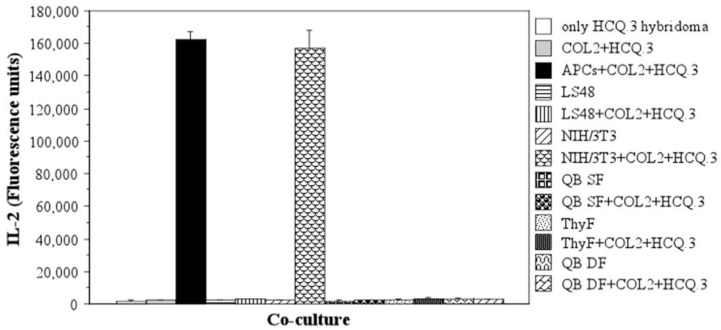
Antigen-presenting capacity of different types of fibroblasts. The ability of LS48, NIH/3T3, synovial fibroblasts (SF), thymic fibroblasts (ThyF), and dermal fibroblasts (DF) to process and present COL2 was evaluated by measuring IL-2 levels in culture medium following 24 h co-culture of fibroblasts with COL2-specific HCQ.3 T-cell hybridoma and 50 µg/mL COL2. The mean of triplicate measurements for each sample is shown.

**Figure 3 ijms-24-10811-f003:**
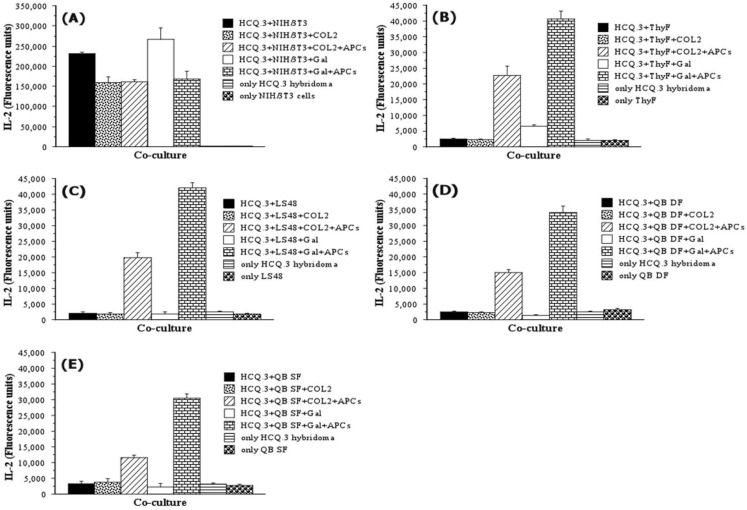
NIH/3T3 fibroblasts activate HCQ.3 cells even in the absence of COL2 antigen. (**A**) NIH/3T3 and HCQ.3 cells were co-cultured for 24 h with or without COL2 and glycosylated COL2_259–273_ peptide, as well as with or without addition of professional APCs. Murine thymic fibroblasts (**B**), LS48 fibroblasts (**C**), QB dermal fibroblasts (**D**), or QB synovial fibroblasts (**E**), and HCQ.3 cells were co-cultured for 24 h with or without COL2 and glycosylated COL2_259–273_ peptide, as well as with or without addition of professional APCs. All samples were assessed in triplicates. Data represent ± SEM.

**Figure 4 ijms-24-10811-f004:**
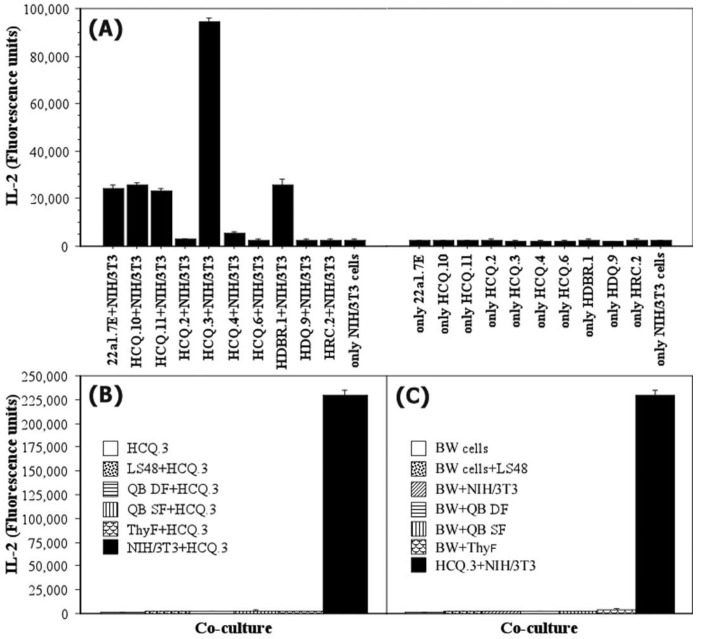
Ability of NIH/3T3 fibroblasts to activate different COL2-specific T-cell hybridomas. (**A**) NIH/3T3 fibroblasts were co-cultured with different COL2-specific hybridomas that recognize either the non-modified COL2_259–273_ epitope or the posttranslationally modified (glycosylated and hydroxylated) immunodominant epitope. Upon 24 h of incubation, IL-2 levels in culture supernatant were measured by enzyme-linked immunosorbent assay (ELISA). (**B**) HCQ.3 hybridoma was co-cultured with different types of fibroblasts—NIH/3T3, LS48, QB DF, QB SF, and ThyF. After 24 h, hybridoma activation was evaluated by measuring IL-2 content in co-culture supernatant. (**C**) Co-culture of BW 5147 cells with NIH/3T3 or LS48, QB DF, QB SF, and ThyF. Following 24 h of incubation, IL-2 levels in culture supernatant were measured by ELISA. All samples were analyzed in triplicate. Data represent ± SEM.

**Figure 5 ijms-24-10811-f005:**
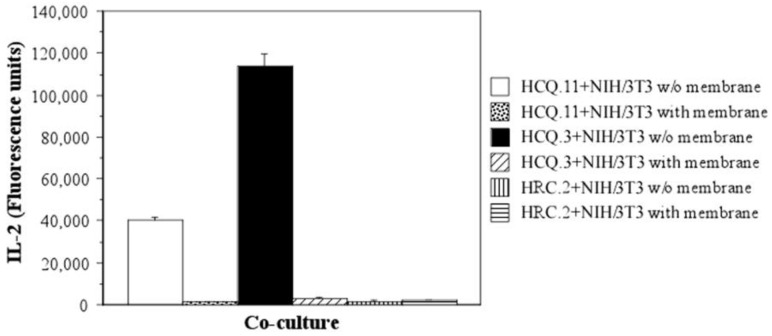
NIH/3T3 fibroblasts interaction with COL2-specific T-cell hybridomas is cell-cell contact-dependent. NIH/3T3 fibroblasts and COL2-specific T-cell hybridomas were co-cultured for 24 h with or without transwell membrane. After 24 h of incubation, IL-2 levels in culture supernatant were measured by ELISA. Data represent ± SEM. All samples were assayed in triplicate.

**Figure 6 ijms-24-10811-f006:**
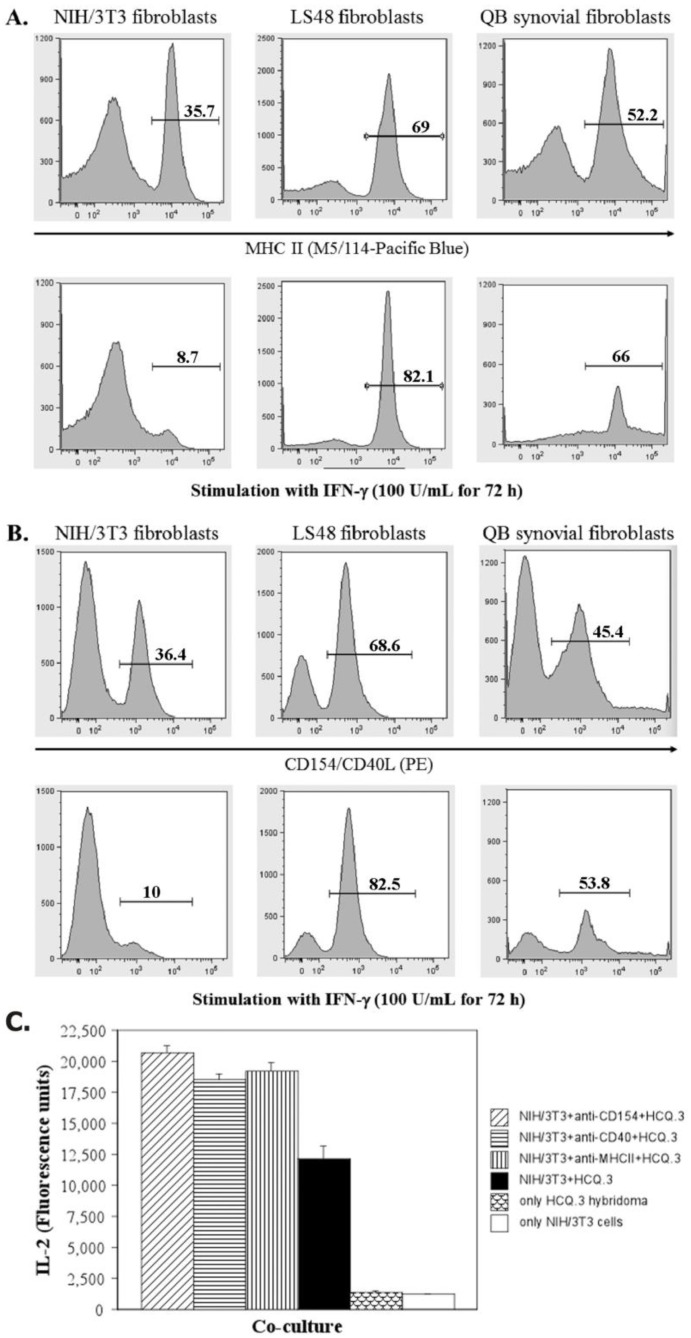
Expression levels of MHCII and CD40L on different types of murine fibroblasts. NIH/3T3, LS48, and QB synovial fibroblasts were cultured in standard complete DMEM medium or in medium with 100 U/mL IFN-γ for 72 h. After that, the cells were trypsinized, stained with fluorochrome-labeled antibodies, and analyzed by flow-cytometry. (**A**) Untreated and IFN-γ-pulsed NIH/3T3, LS48, and QB synovial fibroblasts were analyzed for MHC class II molecules expression using monoclonal MHCII antibody (clone M5/114). (**B**) Expression levels of CD40L on different types of murine fibroblasts following standard culture and INF-γ treatment. (**C**) CD40, CD40L, and MHCII on NIH/3T3 cells were blocked using specific monoclonal antibodies. The fibroblasts were then washed and co-cultured with HCQ.3 T-cell hybridomas. After 24 h of incubation, the IL-2 levels in the culture medium were analyzed by ELISA.

**Figure 7 ijms-24-10811-f007:**
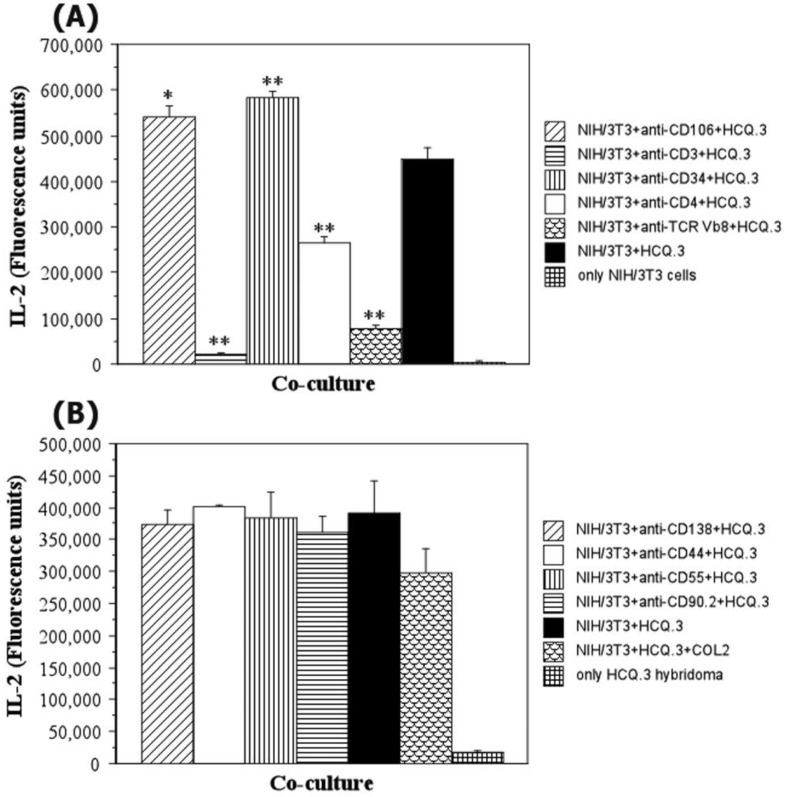
Effect of NIH/3T3 cell surface molecules blocking HCQ.3 activation. (**A**) NIH/3T3 cells were incubated for 1 h with anti-mouse CD106 or CD34 antibodies. HCQ.3 cells were treated with anti-mouse CD33, CD4, or TCR Vβ8 antibodies for 1 h. Thereafter, the cells were washed and co-cultured with untreated HCQ.3 or NIH/3T3 cells. After 24 h of co-culture, T-cell hybridoma activation was analyzed based on IL-2 measurements in culture medium. (**B**) NIH/3T3 fibroblasts were treated with anti-mouse CD44, CD55, CD90.2, or CD138 antibodies for 1 h. After that, the cells were washed and co-cultured with HCQ.3 T-cell hybridomas for 24 h. IL-2 content was measured by ELISA following co-culture. All samples were plated and analyzed in triplicate. Data are shown as ± SEM. * *p* ˂ 0.05, ** *p* ˂ 0.01 vs. NIH/3T3 + HCQ.3.

**Table 1 ijms-24-10811-t001:** T-cell hybridoma reactivities against COL2 protein and COL2_259–273_ peptide with posttranslational modifications. +, reactive; –, no reactivity.

Hybridoma	COL2 Protein	Non-Modified COL2_259–273_ Peptide	COL2_259–273_ Hyl-264 Peptide	COL2_259–273_ Gal-Hyl-264 Peptide	COL2_259–273_ Gal-Hyl-264/270 Peptide
HRC.2	+	+	–	–	–
HDQ.9	+	+	–	–	–
HCQ.4	+	+	–	–	–
HDBR.1	+	–	+	–	–
HCQ.2	+	–	–	+	–
HCQ.3	+	–	–	+	–
HCQ.6	+	–	–	+	–
HCQ.10	+	–	–	+	–
22a1.7E	+	–	–	+	–
HCQ.11	+	–	–	–	+

## Data Availability

Data are contained within the article or available from the corresponding author upon request.

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
