# Peer review of "NIH/3T3 Fibroblasts Selectively Activate T Cells Specific for Posttranslationally Modified Collagen Type II"

_ijms, 2023, doi:10.3390/ijms241310811_

Round 1
Reviewer 1 Report (Previous Reviewer 3)
1) Dear authors, I am happy that you have understood my veiled ( bystander cells) and non veiled suggestions ( verifying if the unusual mechanism exists and can be attributed to fibroblasts in the inflamed state of RA), in order to give strength to some of yours conclusion and better contextualize them in RA, for accept the article.
I understood very well that your aim was to study the involvement of fibroblasts in the presentation of COL II in the phases preceding the inflammation, but since in the first work, as set up on my opinion, your conclusions (fibroblasts may have an unusual way to activate the cells T), were not supported by the data, since ex vivo fibroblasts did not have this unusual activation mechanism, valid only for the NHI/T3 cell line, I had suggested that this mechanism might be existed during inflammation, but not in RA initiation. Moreover, the velated suggestion that the unusual T cells activation may be induced by participation of mimicking antigens and bystander cells, had the aim, one time again, to give substance to your conclusions, that, I repeat, were not supported by the data obtained from ex vivo fibroblasts and to contextualize them in the pathology. Now, Everything was solved as best as possible, by attributing the hypothetical mechanism of activation to other cells that might act as APCs, although the object of the study were fibroblasts
I hope everything is clearer now. The question to investigate in the future will be to verify whether the unusual mechanism of T cell activation by NHI/3T3 is due only to an intrinsic property of the cell line or whether it "really" may belong also in vivo in bystander cells that might active T cells.
2) Line 388 : change” by which fibroblast may be involved “ with “ by which bystander cells, acting as APC” may be involved” as your data don’t’ show that ex vivo fibroblast have this unusual activation mechanism and in line with your reorganized title and paper.
3) line 392: for the same motive, change “ likely this interaction between fibroblast and T cells” with “likely this interaction between NIH/3T3 cells and T cells “
4) Line 414 : change “that RA initiation is possible ” with “that RA initiation might be possible ”…it is better as other experiments need to be performed
5) Line 412: change with “On the other side the fact that till now only NHI/3T3 fibroblast may activate specific COLL 2 T cells……suggest that this unusual mechanism can rather be attributed to bystander cells with participation of mimicking antigens “
6) The second and last clarification is about the question that fibroblasts not process or present COL II protein or derived peptides to T cells, by your data. It is still possible that fibroblasts might present them ,but not as efficiently as their professional counterparts ( figure S2). So Protein and peptide concentration might be important in setting experiments. Also time contact might be important( between hybridoma and fibroblasts ). In my previously comments, my intention was to suggest to introduce, other experiments that might support your conclusion ( fibroblast during RA initiation not present COL II) , repeating the co-culture of fibroblasts and hybridomas with varying these two parameters. Your hybridomas setting, may not to be adapt for co-cultures setting . Now at least, I suggest to introduce this argument into the conclusions also for initiation of RA and not only during joint inflammation . Short stimulus could not give a complete view of the behavior of the cells, as also written by you in the first paper (line 317-320) and in the last version (348-351).
7) Line 128: Anyway please introduce recall of literature: number 53 and 81
8) Fig 3: missing data, please introduce data also for QBSF, it is important as SF are involved in RA
9) Line 271: introduce fig 7 as Fig 7A and 7B
10) Line 195 : introduce … “with ( fig S1) and without (Fig 4B)”
good
Author Response
Please see the attachment.

Reviewer 2 Report (New Reviewer)
Dear authors,
The paper submittted by Dzhambazov et al. To the prestigious journal IJMS approaches a nice topic and I belive that it might be of considerable interest for the readers.
I hope that my remarks will be useful in order to increase the quality of the paper:
1. Usually the corresponding author is the senior (last) author as a coordinator of the project.
2. Introduction. (Line 32) It would be nice to mention the role of fibroblast in periodontal disease (PD).
3. Discussion. (Line 288) - RA is a systemic affection that is extremely well correlated to periodontal disease so emphasizing this aspect in the Discussion section should be higly recommended.
4. Discussion. (Line 288) - If authors consider appropiate they might take into consideration this article for improving the Discussion section regarding the connexion between RA and PD. https://doi.org/10.3390/bioengineering10010061
5. Line 566 – The statistical method should be describet more accurately.
6. Line 570 – My suggestion is to improve the Conclusions section in such a way that it will reflect the clinical/practical impact of the results in improving current complex protocol of treating RA in a comprehensive manner.
Overall, the paper is well written, the topic is of interest and it fits the scope of the journal.
Best of luck of your work!
Author Response
Please see the attachment.

This manuscript is a resubmission of an earlier submission. The following is a list of the peer review reports and author responses from that submission.
Round 1
Reviewer 1 Report
Summary:
Rheumatoid arthritis (RA) is a chronic autoimmune and inflammatory disease that mostly affects joints. Synovial fibroblasts (SF) play a key role in the pathogenesis of RA as main contributors to the inflammatory process, and a self-antigen associated with the inflammatory process is type II collagen (COL2). To examine the ability of different types of fibroblasts to activate COL2-specific T-cell hybridomas by processing and presenting COL2, Balik Dzhambazov, et al cultured T-cell hybridoma HCQ.3 with synovial fibroblasts, thymic fibroblasts, destructive LS48 fibroblasts, and noninvasive NIH/3T3 fibroblasts respectively. The data unravel that only NIH/3T3 cells can activate COL2-specific T-cell hybridomas, more interestingly, even in the absence of COL2 or posttranslationally modified COL2 peptide.
General concept comments:
(1) Insufficient introduction of the functions of COL2 in bone joints and of the roles of COL2 in the pathogenic process of RA.
(2) Insufficient introduction of roles of MHCII in antigen presentation during the pathogenic process of RA, since the authors used an entire figure (Figure 1) to show the expression of Aq in each type of fibroblasts.
(3) The data show that NIH/3T3 fibroblasts, but not synovial fibroblasts, are able to process and present COL2 to COL2-specific T-cell hybridomas. Please discuss the relevance and significance of the findings in this manuscript to the pathogenesis of RA.
Specific comments:
Missing/incorrect controls. The authors used the quantification of IL-2 fluorescence units to represent the activation of T-cell hybridoma HCQ.3. Thus, in these experiments, HCQ.3 should not be a variable. For example, in Figure 2, the control should be “only HCQ.3”, “CII+HCQ.3”, and “each fibroblastic type+HCQ.3 (w/o CII)”, and similar missing/incorrect controls in Figure 4A. Without appropriate controls, these data are nonsensical.
Reviewer 2 Report
Dear Authors,
Please improve and extent your introduction
Please add more conclusion to your article
Kind regards
Dear Authors,
Minor editing of English language required
Kind regards
Reviewer 3 Report
line 76: Why the authors don't consider to analyse also different fibroblasts from a mouse model of RA? It would have been interesting
line 90: show expression of Aq also from synovial/dermal fibroblast by microscopic analysis, if you have
line 131: It is usefull to check also with LS48 and QB fibroblasts
line 142: it could be usefull to check and to confirm that mouse fibroblasts (synovial dermal and thymic ) are unable to active also other different T cell hybridoma clones with and without COL2 antigens.
Fig 4C: missing data BW + QB
Finally as said by authors "the results gained from their experiments skewed the course of their investigatiom from examination of APC capacity of different fibroblast types ( I to show if fibroblasts may have a role in RA as APC for COL2) to study the unusula antigen-independent mechanism of T cell hybridoma activation in co-culture with NIH/3T3. The question is ...which type of interest can I have the authors's preliminary data, to deepen, in the field of rheumatoid arthritis? Moreover the capacity to stimule the HCQ.3 hydridoma and the other Hybridoma clones "seems" to include only NIH/3T3, a cell line, but neither different normal mouse fibroblasts (stimulated and not stimullated) nor the fibroblastic destructive line LS48. Try to better contextualize your data in RA ( rheumatoid arthritis), if possible
In addition, in the conclusion the authors say that different types of murine fibroblasts are not able to process COL2 and present COL2 peptides to arthoritogenic T cells but these affermation must be proven with other experiments for example analyzing also fibroblasts from a mouse model of RA.
Round 2
Reviewer 1 Report
The authors addressed all of my questions and concerns in their point-by-point response to my comments. I believe the manuscript has been sufficiently improved to warrant publication in IJMS.
Reviewer 3 Report
1) line 76: Why the authors don't consider to analyse also different fibroblasts from a mouse model of RA? It would have been interesting
The aim of our work was to evaluate the capacity of different types of fibroblasts to process and present COL2 as a main structural compound of the cartilage, and whether the fibroblasts may be involved in the initiation of the RA pathogenesis, similarly to the professional APCs. This was the reason to use fibroblasts that are not affected by inflammatory processes. Several authors (cited in the manuscript) have shown that synovial fibroblasts from inflamed joints are able to present antigens. However, these are already activated fibroblasts by other signaling pathways. And may be they are already loaded with antigens.
I think that to use fibroblasts by RA mouse model with COLL 2 protein and with COLL2 peptides in coculture with Hybridomas, could be will be useful to verify if there is activation and if, in an inflammation situation as RA, fibroblasts are able to process COLL2 or to be stimulated by Coll2 peptides, presents for example in synovial fluids. To discerner if these fibroblasts are activated by others signaling pathway it is sufficient to use as control the same fibroblasts with hybridomas and without CoLL2 protein/peptides or use scramble peptides to eventually notice an additive effect, produced by Coll 2 or Coll2 peptide. Anyway if you want to study “the fibroblasts and verify if they may be involved in the initiation of the RA pathogenesis, similarly to the professional APCs”, please clarify better or add this phase into introduction.
2) line 90: show expression of A q also from synovial/dermal fibroblast by microscopic analysis, if you have.
Since the fibroblasts were isolated from mice with Aq haplotypes (genetically determined) we thought that it is not necessary to prove the expression of Aq. An additional figure (Figure S1) was now provided in the supplementary material.
OK
3) line 131: It is useful to check also with LS48 and QB fibroblasts.
These data were added to the manuscript (Figure 3)
OK
4) line 142: it could be useful to check and to confirm that mouse fibroblasts (synovial dermal and thymic) are unable to active also other different T cell hybridoma clones with and without COL2 antigens.
We provided these data from the performed hybridoma tests as supplementary materials (Figure S2).
I mean to show also antigen presenting capacity stimulation not only with COLL2 but also with non modified epitope or the post translationally modified, glycosylated and hydroxylated immunodominant epitopes, us done with NIH/3T3 fibroblasts. I don’t see these experiments and I don’t see also before, a dose dependent and time course experiments with COLL2 and with antigens. It was reported by literature that FLS require prolonger incubation with antigen and longer coculture with T cells/hybridoma . This suggests that that FLS might not process or present antigen as efficiently as their professional counterparts or they might use different mechanisms ( figure S2). Protein and antigen concentration may be also influence fibroblast ability to present.
5) Fig 4C: missing data BW + QB.
Three of the figures (including Figure 4) were reorganized adding more controls.
FIG 4A : missing HCQ.4
6) Finally as said by authors "the results gained from their experiments skewed the course of their investigation from examination of APC capacity of different fibroblast types ( I to show if fibroblasts may have a role in RA as APC for COL2) to study the unusual antigen-independent mechanism of T cell hybridoma activation in co-culture with NIH/3T3. The question is ...which type of interest can I have the author’s preliminary data, to deepen, in the field of rheumatoid arthritis? Moreover the capacity to stimulate the HCQ.3 hybridoma and the other Hybridoma clones "seems" to include only NIH/3T3, a cell line, but neither different normal mouse fibroblasts (stimulated and not stimulated) nor the fibroblastic destructive line LS48. Try to better contextualize your data in RA ( rheumatoid arthritis), if possible.
We have revised some of the sentences in the text showing clearly that fibroblasts are not able to process COL2 as APCs, but there is another contact-dependent mechanism, which could play a role in the RA pathogenesis.
See general consideration
7) In addition, in the conclusion the authors say that different types of murine fibroblasts are not able to process COL2 and present COL2 peptides to arthritogenic T cells but these affirmations must be proven with other experiments for example analyzing also fibroblasts from a mouse model of RA.
As we discussed above, this will be another situation, and probably synovial fibroblasts will be able to present some short antigens (peptides), but from our data is clear that fibroblasts are not able to process COL2 protein
To be confirmed by other experiments (see point 4)
General consideration:
The authors have partially integrated the requested one but two major perplexities remain:
1) The authors suggest the existence of another mechanism by which fibroblasts may be involved in the pathogenesis of RA, namely through direct contact to arthritogenic T cells activating them by using molecules that mimic the relevant antigens, but the fact that this situation occurs only with the fibroblast cell line NHI73T3, but neither in ex vivo mouse fibroblasts nor in fibroblast cell line LS48 , rather could also suggest an artifact not of the hybridoma, as demonstrated by the authors, but an artifact due to intrinsic characteristics of the NHI/3T3 cell line. The authors should also consider this possibility.
The authors’ hypothesis (“to verify if fibroblasts may be involved in the initiation of the RA pathogenesis”) and conclusions, (“Our findings suggest the existence of another mechanism by which fibroblasts may be involved in the pathogenesis of RA, namely through direct contact to arthritogenic T cells activating them by using molecules that mimic the relevant antigens”) are not validated by their data as the results are obtained only for NHI/3T3 fibroblasts but not by ex vivo fibroblasts . The authors don’t express any opinion about this.
2) Scientific data supporting authors’ conclusion perhaps could be obtained by analyzing fibroblasts from mouse models of RA ex vivo. These fibroblasts could have a behavior similar to that noted for NHI /3T3 fibroblasts, induced by the inflammatory processes present in RA disease, or by bystander cells that could induces or activate the expression of molecules capable of mimicking relevant antigens, but the authors did not analyzed this situation. These fibroblasts may be also analyzed in presence of IFN ϒ or inflammatory cytokines
In conclusion, on my opinion, data don’t support sufficiently the authors’ conclusions and the existence of an in vivo unusual mechanism by which fibroblast are able active T cells. More scientific data / conditions should be analyzed to support author’s conclusions about on a side a real and unusual mechanism of T cells activation by some fibroblasts and , on the other side, to confirm the inability of fibroblasts to present COLII PROTEIN/ ANTIGENS. The authors moreover don’t speak about neither about limitations of their studies nor of future aims in their work.
Many factors is necessary to take in account to study the possibility that fibroblast may act as APC in vivo, ex vivo and in vitro. Some of these are: the presence of an inflammatory milieu, the presence of bystander cells in inflammation place, MHC II restriction of T cells/hybridoma (the correct MHCII subtype), experimental condition ( concentration and duration time of stimulus).